# Observation of anomalous amplitude modes in the kagome metal CsV$_3$Sb$_5$

Gan Liu[1], Xinran Ma[1], Kuanyu He[1], Qing Li [1], Hengxin Tan[2], Yizhou Liu [2], Jie Xu[1], Wenna Tang[1], Kenji Watanabe [3], Takashi Taniguchi [4], Libo Gao [1,5], Yaomin Dai [1,5], Hai-Hu Wen [1,5], Binghai Yan [2✉] & Xiaoxiang Xi [1,5✉]

The kagome lattice provides a fertile platform to explore novel symmetry-breaking states. Charge-density wave (CDW) instabilities have been recently discovered in a new kagome metal family, commonly considered to arise from Fermi-surface instabilities. Here we report the observation of Raman-active CDW amplitude modes in CsV$_3$Sb$_5$, which are collective excitations typically thought to emerge out of frozen soft phonons, although phonon softening is elusive experimentally. The amplitude modes strongly hybridize with other superlattice modes, imparting them with clear temperature-dependent frequency shift and broadening, rarely seen in other known CDW materials. Both the mode mixing and the large amplitude mode frequencies suggest that the CDW exhibits the character of strong electron-phonon coupling, a regime in which phonon softening can cease to exist. Our work highlights the importance of the lattice degree of freedom in the CDW formation and points to the complex nature of the mechanism.

[1] National Laboratory of Solid State Microstructures and Department of Physics, Nanjing University, Nanjing 210093, China. [2] Department of Condensed Matter Physics, Weizmann Institute of Science, Rehovot 7610001, Israel. [3] Research Center for Functional Materials, National Institute for Materials Science, 1-1 Namiki, Tsukuba 305-0044, Japan. [4] International Center for Materials Nanoarchitectonics, National Institute for Materials Science, 1-1 Namiki, Tsukuba 305-0044, Japan. [5] Collaborative Innovation Center of Advanced Microstructures, Nanjing University, Nanjing 210093, China. ✉email: binghai.yan@weizmann.ac.il; xxi@nju.edu.cn

**M**aterials with a kagome lattice can host rich phenomena encompassing quantum magnetism[1,2], Dirac fermions[3,4], nontrivial topology[5–7], density waves, and superconductivity[8–10]. The recently discovered kagome metals $A$V$_3$Sb$_5$ ($A$ = K, Rb, or Cs)[11,12] offer a new platform to study the interplay of these phenomena. These compounds have Fermi levels close to Dirac points or van Hove singularities[12–14], leading to a plethora of possible intriguing ground states. Indeed, charge-density waves and superconductivity have been discovered[12,15,16], with ample evidence showing that both types of orders are exotic. For example, the CDW transition is accompanied possibly by a large anomalous Hall effect[17,18], and the superconductivity features a pair-density wave state[19].

The nature of the CDW state and the mechanism for its formation have been under close scrutiny. In this work, we focus on CsV$_3$Sb$_5$, which has a CDW transition temperature $T_{CDW}$ = 94 K[12]. Both hard-X-ray and neutron scattering showed the lack of soft phonons (phonon modes that show frequency softening upon cooling toward a phase transition)[20,21], although density functional theory (DFT) calculations found two phonon instabilities at the $M$ and $L$ points of the Brillouin zone[22–24]. Considering that a 2 × 2 modulation of the crystal lattice is well established[12,19,25–27], the absence of soft modes apparently breaks a pattern proven generic to many known CDW systems—a soft phonon freezes to zero frequency and triggers the formation of a distorted lattice[28]. Currently, there is still no consensus on the form of the in-plane structure and the $c$-axis periodicity[27,29]. The roles of Fermi-surface nesting and electron–phonon coupling are also debated. Because the period of the 2 × 2 superlattice matches perfectly with the Fermiology of the van Hove singularity, it is natural to ascribe the CDW transition to Fermi surface nesting[22,30–32], supported by the appreciable partial gapping of the Fermi surface observed experimentally[33–37]. However, the calculated electronic susceptibility lacks the expected divergence[38,39], and the effect of electron–phonon coupling may not be dismissed[21,39].

Because the CDW features lattice distortions, studies of the lattice degree of freedom can offer insight into the mechanism. Raman scattering is a valuable tool in this respect. In well-studied CDW systems, such as the transition metal dichalcogenides, as a soft phonon mode condenses to form a distorted lattice, new Raman-active collective excitations, known as amplitude modes, emerge, providing a direct probe of the CDW order parameter[40–42] (see Fig. 1a). Conversely, the observation of amplitude modes is typically considered as evidence for the soft mode. The temperature dependence of the amplitude modes as well as the zone-folded modes, which become Raman-active due to zone folding induced by the superlattice, can both reflect the CDW transition[43–49]. Combined with symmetry information from polarization-resolved measurements, constraints can be set on the possible CDW ground state.

Here, we report Raman scattering measurements on CsV$_3$Sb$_5$. We observe a multitude of CDW-induced modes, whose symmetries and frequencies are in good agreement with DFT calculations for a single-layer CsV$_3$Sb$_5$ under inverse Star of David distortion. The observed temperature dependence of these modes and their calculated evolutions with varying lattice distortion allow us to identify two of them as amplitude modes, emerging from the predicted soft modes, although the soft modes are elusive experimentally. In contrast to mostly independent amplitude modes and zone-folded modes in well-known CDW materials[44–46], we show that they hybridize strongly in CsV$_3$Sb$_5$, causing spectral weight redistribution to the latter and rendering them amplitude-mode-like. The anomalous hybridization and the large values of the amplitude mode frequencies provide evidence of strong-coupling CDW, offering a possible explanation for the lack of soft modes. These results stress the importance of the lattice degree of freedom and electron–phonon coupling in the CDW formation in CsV$_3$Sb$_5$.

## Results

**Raman-active modes in CsV$_3$Sb$_5$.** CsV$_3$Sb$_5$ crystallizes in a hexagonal lattice with the $P6/mmm$ space group[11]. Figure 1b shows the unit cell of its crystal structure. The V atoms form a kagome net interspersed by Sb atoms (labeled Sb1), all within the $ab$-plane. The V atoms are further bonded by Sb atoms above and below the kagome plane (labeled Sb2). These V$_3$Sb$_5$ slabs are separated by Cs layers, with weak coupling between them to form a quasi-two-dimensional (quasi-2D) structure. Factor group analysis yields three Raman-active phonon modes, $\Gamma_{Raman}$ = $A_{1g} + E_{2g} + E_{1g}$. The former two can be detected when the photons are polarized in the $ab$-plane, satisfied by the back-scattering geometry used in our experiment. These intense modes are marked by dashed lines in Fig. 1c, d. They involve only the Sb2 atoms, with their atomic vibrations along the $c$-axis and within the $ab$-plane for the $A_{1g}$ and $E_{2g}$ modes, respectively; see Fig. 1b. The $E_{2g}$ modes are a pair of degenerate vibrations with opposite circling directions, i.e., opposite chiralities (see Supplementary Note 1). These two types of symmetries can be distinguished by polarization-resolved measurements. Specifically, the $A_{1g}$ modes can be detected in the XX and LL polarization configurations, whereas the $E_{2g}$ modes appear in the XX, XY, and LR configurations. Here, XX and XY represent collinear and cross-linear polarization for the incident and scattered photons, and LL and LR involve circularly polarized light with left (L) and right (R) helicity. A comparison of data in all four configurations is included in Supplementary Fig. 1.

Below $T_{CDW}$, multiple peaks emerge, highlighted by the dotted lines in Fig. 1c, d. Their origin will be discussed in the next sections. These modes are rather weak compared to the main lattice phonons. Their disappearance at 100 K suggests a close correlation with CDW formation. In contrast, many weak peak-like structures below 100 cm$^{-1}$ lack temperature dependence, whose origin is unclear.

Figure 1e compares the observed Raman mode frequencies with those from DFT calculations for a single layer of CsV$_3$Sb$_5$[22], considering two possible forms of lattice distortion, the Star of David (SD) and inverse Star of David (ISD, also referred to as trihexagonal) structures. Both of them show the same number of $A_{1g}$ and $E_{2g}$ modes, but with different ordering. Overall, the calculated ISD phonons agree much better with the experimental results, as shown in the figure and in Supplementary Tab. 1. All the five predicted $A_{1g}$ modes and five out of the eight predicted $E_{2g}$ modes are observed. The observed $A_{1g}$ mode below 50 cm$^{-1}$ is unaccounted for by our calculations. This mode was also observed by pump-probe time-resolved spectroscopy, which, when compared with calculations taking into account interlayer coupling, was assigned as a Cs-mode due to CDW modulation along the $c$-axis[23]. Except for this mode and the three missing $E_{2g}$ modes due to their weak scattering cross section, the symmetry ordering of all the other modes is in exact agreement between the experiment and theory. These results suggest that the CDW ground state consists of weakly coupled layers dominated by ISD-type distortion, but CDW modulation along the $c$-axis is also indispensable. Since the single-layer CsV$_3$Sb$_5$ holds the key to unraveling the CDW mechanism, we attempted creating atomically thin CsV$_3$Sb$_5$ by mechanical exfoliation. However, the loss of crystallinity impeded further investigation (Supplementary Figs. 2 and 3). The almost non-detection of modes folded from the $L$-point may be attributed to weak interlayer interaction, because the $M$- and $L$-point instabilities differ only in the

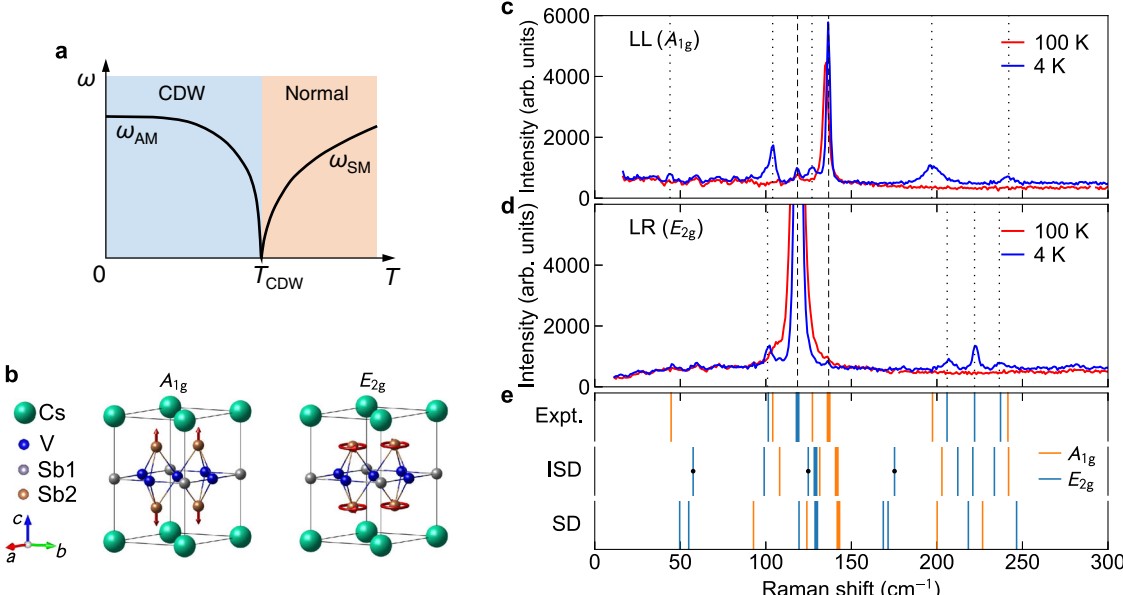

**Fig. 1 Raman-active phonon modes in CsV₃Sb₅. a** Schematic illustration of the relation between the soft mode and amplitude mode in typical CDW materials, showing the latter emerges after the former freezes below $T_{CDW}$. $\omega_{AM}$: amplitude mode frequency. $\omega_{SM}$: soft mode frequency. **b** Crystal structure of $CsV_3Sb_5$. Sb sites with different Wyckoff positions are labeled as Sb1 and Sb2. The arrows illustrate the vibration patterns of the main lattice $A_{1g}$ and $E_{2g}$ modes. The $E_{2g}$ mode is doubly degenerate, and only one form is shown. **c, d** Raman spectra measured on the *ab*-plane at 100 K and 4 K in the LL and LR polarization configurations. The dashed lines denote the main lattice phonons, and the dotted lines indicate the CDW-induced modes. **e** Comparison of the measured (Expt.) and the calculated Raman mode frequencies for the inverse Star of David (ISD) and Star of David (SD) lattice distortions. The thick lines denote the main lattice phonons. The dots indicate modes undetected in our experiment.

interlayer ordering. Indeed, we have considered various forms of *c*-axis modulation, and all of them predict a large number of Raman modes far exceeding that observed experimentally (Supplementary Note 2). Polarization-angle dependent measurements (Supplementary Fig. 4) show that either the *c*-axis modulation is too weak to induce clear anisotropic Raman response, or those candidate stacking orders with the $D_{2h}$ point group can be ruled out.

**Temperature dependence of Raman modes.** Figure 2a, b shows the temperature-dependent Raman intensity color plot for $CsV_3Sb_5$, obtained in the LL and LR configurations, respectively. The intense $A_{1g}$ and $E_{2g}$ main lattice modes are the most conspicuous features. Figure 2e–g shows the frequency (with the corresponding value at 200 K subtracted), linewidth (full width at half maximum), and normalized integrated area for both modes, extracted from Lorentzian fits of the peaks. The $A_{1g}$ frequency sharply increases below $T_{CDW}$, whereas the $E_{2g}$ frequency exhibits a subtle kink across the CDW transition. This is consistent with the planar ISD lattice distortion mainly involving V atoms, forcing the Sb2 atoms to displace along the *c*-axis, hence affecting the out-of-plane vibration of the $A_{1g}$ mode more effectively. The calculated phonon vibration patterns and frequencies confirm this picture (see Supplementary Fig. 5 and Supplementary Table 1). The CDW transition also causes a faster decrease in the linewidths below $T_{CDW}$. This can be understood as being due to the CDW-induced partial gapping of the Fermi surface[33–37], which reduces the electron–phonon interaction. The integrated peak intensity for both phonons increases upon warming, in line with increased thermal phonon populations. The rate of increase is faster when approaching $T_{CDW}$ from below, and interestingly, the value saturates below approximately 50 K. The renormalization of the phonon parameters across the CDW transition evidences sizable electron–phonon coupling.

CDW-induced modes are labeled in Fig. 2a, b. Except for the $A_1$ mode, there appears to be two types of modes, represented by $A_2$ and $E_3$. $A_2$ exhibits appreciable softening and broadening upon warming toward $T_{CDW}$. It is overdamped before disappearing, visualized in the color plot in Fig. 2a as the streak of signal below $100\,cm^{-1}$ between 60 and 90 K (see also Supplementary Fig. 6). These are signatures of a CDW amplitude mode[43–47], caused by the collapse of coherent CDW order near $T_{CDW}$. $E_3$ shows a smaller change of frequency and much less broadening, more consistent with the characteristics of a zone-folded mode[44], as this type of mode arises from folding a zone-boundary phonon to the zone center, making its temperature dependence of the frequency as weak as that of normal phonons. Figure 2c, d compares the distinct temperature dependence of these two types of modes. While $A_2$ broadens significantly above 40 K, $E_3$ maintains its linewidth and suddenly vanishes above ~80 K. The dramatic difference in the linewidth broadening is quantified in Fig. 2i. Figure 2h shows the frequencies of all the observed Raman modes on the same scale. Upon warming, the CDW-induced modes ($A_1$ excluded) soften more dramatically than the main lattice modes. While it is tempting to assign most of them as amplitude modes because of the apparent softening behavior, we show below that they are in fact zone-folded modes, mixed with the amplitude modes to partially inherit their properties.

**Nature of CDW-induced modes.** Although soft phonons were not detected experimentally, our DFT results show that the formation of CDW in $CsV_3Sb_5$ is similar to that in other well-known systems[43–47], in the sense that a soft phonon mode at the CDW wavevector condenses and gives rise to a distorted lattice[41]. The imaginary phonon modes of pristine $CsV_3Sb_5$ at three $M$ points (see Supplementary Fig. 7) transform as irreducible representation $M_1^+$ ($A_g$) of the space group $P6/mmm$ (little co-group $D_{2h}$). Figure 3a shows that they form triply degenerate modes at $\Gamma$ due to the artificial band folding without lattice distortion, in which these modes are not

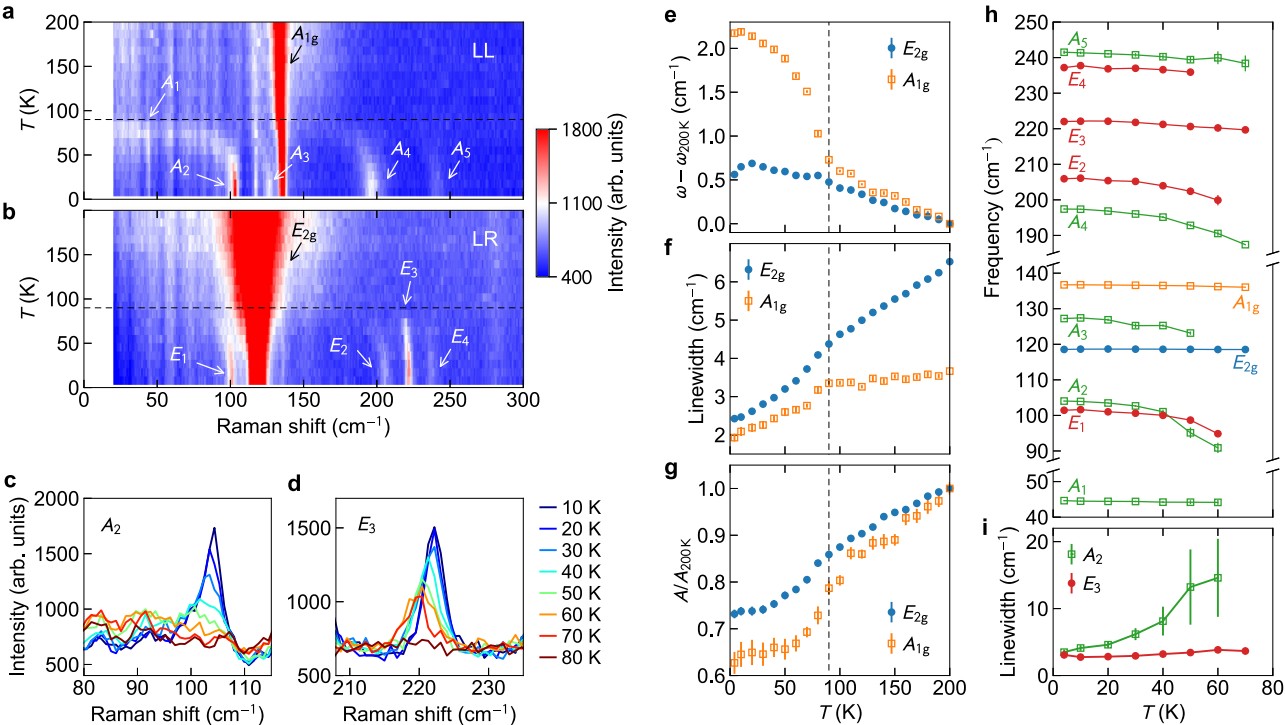

**Fig. 2 Evolution of the Raman modes in CsV₃Sb₅ across the CDW transition.** **a**, **b** Temperature-dependent Raman intensity color plot for $CsV_3Sb_5$, measured in the LL and LR configurations. The normal phonon modes are labeled in black and the CDW-induced modes in white. The dashed lines mark $T_{CDW}$. **c**, **d** Temperature-dependent spectra for the $A_2$ and $E_3$ modes. **e**–**g** Frequency, linewidth, and amplitude for the $E_{2g}$ and $A_{1g}$ main lattice phonons. The frequency and amplitude are compared to the corresponding values at 200 K. **h** Temperature dependence of the Raman mode frequencies. **i** Temperature dependence of the linewidth of the $A_2$ and $E_3$ modes. Error bars are standard deviations obtained from the least-squares fits to the phonon peaks.

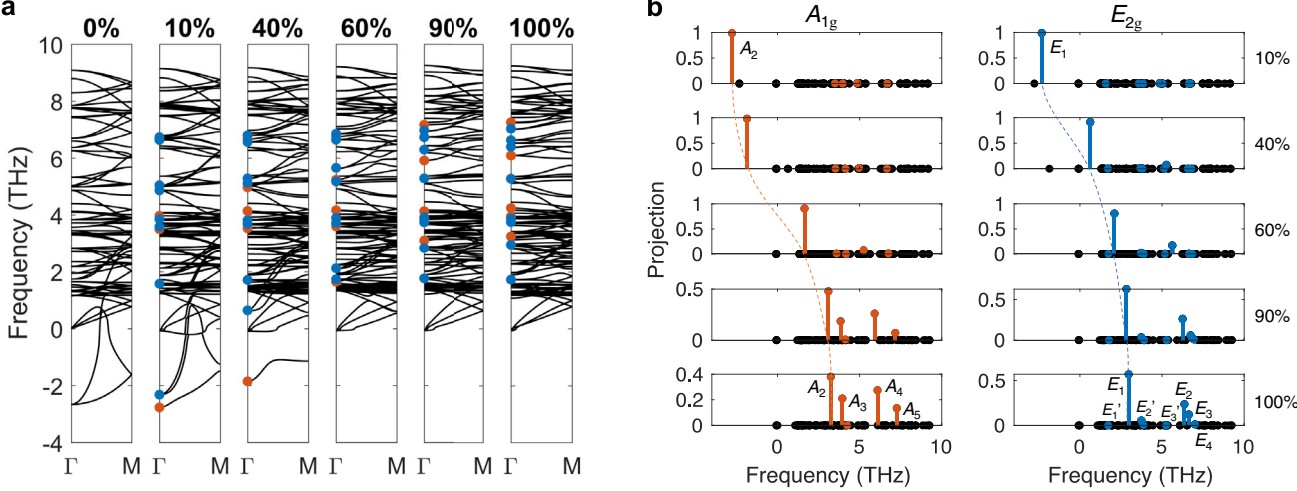

**Fig. 3 Phonon band structures and mode mixing in the process of CDW distortion.** **a** Phonon band structures directly calculated by DFT. Here, 100% (0%) refers to the fully stable ISD ($2 \times 2$ pristine) structure. 10% refers to the intermediate structure with 10% distortion from the pristine to ISD phases. After $2 \times 2 \times 1$ band folding with no distortion, three imaginary modes ($M_1^+$) are folded to Γ. A weak ISD-type distortion lifts the degeneracy and leads to $A_{1g}$ and $E_{2g}$ modes. The ISD distortion gradually transforms imaginary modes to real. **b** Projections of the imaginary $A_{1g}$ ($E_{2g}$) mode with 10% distortion to all the other phonon modes at Γ, as evolving into the stable ISD phase (100%). We highlight all $A_{1g}$ and $E_{2g}$ modes by orange and blue dots, respectively, at the Γ point. The dashed orange (blue) curve in (**b**) guides eyes to show the evolution of the imaginary $A_2$ ($E_1$) modes in the CDW distortion.

measurable in Raman. Only when CDW appears, they come out as amplitude modes, characterizing the CDW transition. With CDW distortion, they decompose to a singlet $A_{1g}$ mode and a doublet $E_{2g}$ mode under the point group $D_{6h}$ (see Supplementary Note 3):

$$3M_1^+ \rightarrow A_{1g} \oplus E_{2g}. \qquad (1)$$

Despite that intermediate structures in Fig. 3 are unstable structures with finite atomic forces, calculated force constants are still valid in the harmonic approximation (see "Methods"). Corresponding pseudo-phonon bands can provide useful insights to understand the soft mode evolution with respect to CDW distortion. As the lattice distorts from the pristine phase to the

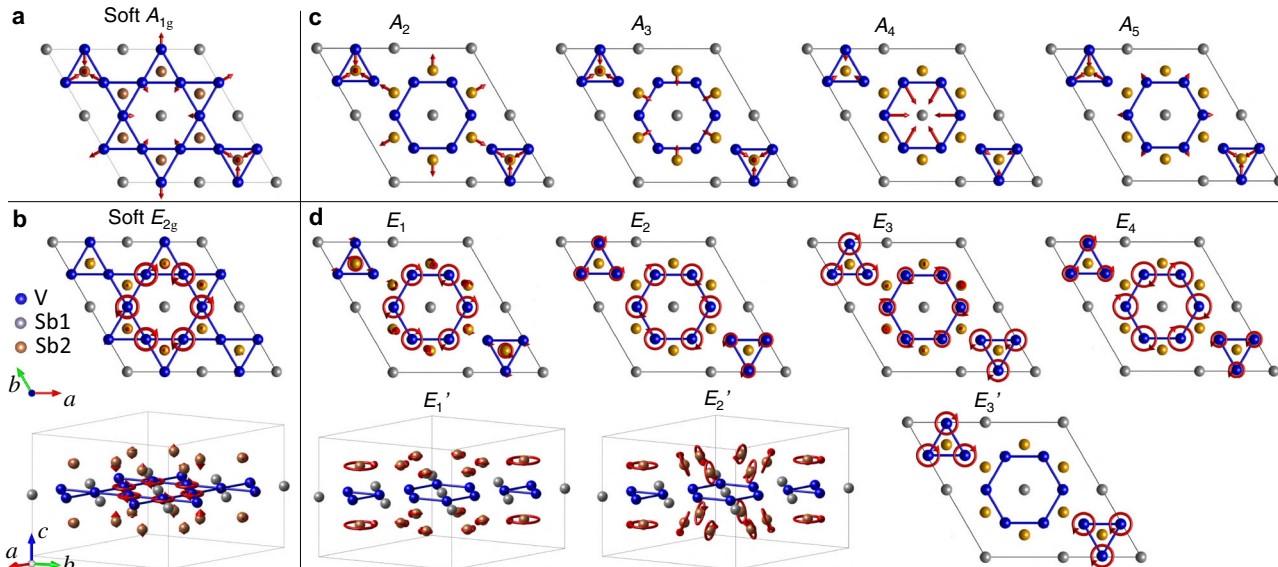

**Fig. 4 Real space displacement patterns of the imaginary soft modes and the stable CDW-induced Raman modes in the 2 × 2 × 1 ISD phase. a, b** Soft modes with $A_{1g}$ and $E_{2g}$ symmetries, respectively. **c, d** CDW-induced $A_{1g}$ and $E_{2g}$ stable modes. The $E_{2g}$ modes are pairs of chiral phonons and only one chiral mode is shown. The radius of the circles represents the amplitude of the vibration, and the arrow on the circles stands for the initial phase of the vibration. Cs atoms are omitted in the crystal structure for clarity, because they do not contribute to lattice vibrations.

stable ISD pattern, the imaginary $A_{1g}$ and $E_{2g}$ modes turn real with positive frequencies, expected to be observable as two Raman-active amplitude modes. The phonon displacement patterns of the soft $A_{1g}$ and $E_{2g}$ modes are shown in Fig. 4a, b, dominated by vibrations of V atoms. The $A_{1g}$ mode is fully symmetric, involving breathing-type motion for the V triangles, V hexagons, and Sb2 atoms. The $E_{2g}$ mode involves circling motion for the atoms forming the V hexagons, while the amplitude for the Sb2 vibration is almost ten times smaller.

DFT calculations further reveal that the amplitude modes strongly hybridize with the other CDW-induced Raman modes (i.e., the zone-folded modes always at positive frequencies at the Γ point in Fig. 3a), rendering them amplitude-mode-like, hence their apparent temperature-dependent frequencies. Figure 4c, d shows the real space displacement patterns of all the CDW-induced modes in the 2 × 2 × 1 ISD phase. $A_{2-5}$ and $E_{1-4}$ correspond to those in Fig. 2, and $E'_{1-3}$ are undetected experimentally. Comparison with Fig. 4a, b shows that $A_{2-5}$ ($E_{1-4}$) all resemble the $A_{1g}$ ($E_{2g}$) soft mode, but the difference is also apparent. The similarity results from hybridization of phonon modes.

To quantify the mode mixing, we calculated the overlap between the soft modes and all the real modes of the stable ISD phase by projecting the phonon dynamical matrix eigenvectors, $P_f = |\langle \mathbf{u}_f | \mathbf{u}_{SM} \rangle|^2$, where $|\mathbf{u}_{SM}\rangle$ refers to the eigenvector of the soft modes shown in Fig. 4a, b and $|\mathbf{u}_f\rangle$ refers to the eigenvector of the mode at frequency $f$ in the ISD phase. The results in Fig. 3b show that as the soft $A_{1g}$ and $E_{2g}$ modes shift from negative to positive frequencies and turn into amplitude modes, they hybridize with most of the zone-folded modes belonging to the same irreducible representation. The amount of calculated projection in the stable ISD phase correlates reasonably well with the observed mode intensity in Fig. 1c, d. $A_2$ and $E_1$ are residual amplitude modes after mode mixing. $E'_{1-3}$ show minor projection from the $E_{2g}$ soft mode because of negligible eigenvector overlap, and accordingly their scattering cross section is weak. $E_{2-4}$ all involve V triangles (Fig. 4d), indicating that they have contributions unrelated with the $E_{2g}$ soft mode. Indeed, as discussed earlier, $E_3$ shows clear experimental signatures of a zone-folded mode.

The $A_2$ mode was also observed by Wulferding et al. in their Raman study[50] and by time-resolved pump-probe spectroscopy[23,51]. However, in these works, it was suggested to emerge below ~60 K[23,50,51], hence ascribed to another phase transition associated with a unidirectional order[19,25–27]. According to our data (Fig. 2a), the $A_2$ mode survives above 60 K, and there is no clear evidence for two distinct phase transitions. Its vibration pattern shown in Fig. 4c confirms no relation with the unidirectional order. Raman scattering, as a bulk probe, is probably not sensitive enough to the unidirectional order, due to its possible surface origin[26,52] and its existence in nanoscale domains[26]. A second bulk transition well below $T_{CDW}$ was recently revealed by multiple techniques[53–55], which evaded detection by our Raman measurements, possibly also due to the lack of sufficient sensitivity. Another Raman study by Wu et al.[56] reported a similar set of modes as ours, but with different relative intensities. They also observed extra modes that are possibly due to stronger c-axis modulation in their sample.

## Discussion

The anomalously large hybridization between the amplitude modes and zone-folded modes is rare, because they are mostly decoupled in the canonical CDW materials, with the amplitude modes dominating the spectral intensity[44–46]. The hybridization is highly unusual, because the $A_2$ and $E_1$ amplitude modes and the zone-folded Raman modes span a wide frequency range, and they do not overlap in energy (except for $E_1$ and $E'_1$) to exhibit the typical anti-crossing[57,58]. Their strong coupling suggests that the hybridization occurs indirectly, through interaction with the common electronic system. As the Fermi surface instability associated with the van Hove singularity is from the V bands[22], modes mainly involving V (including $A_{2-5}$ and $E_{1-4}$) naturally mix with the amplitude modes, whereas those mainly involving Sb (including $E'_{1-2}$ and the $A_{1g}$ and $E_{2g}$ main lattice modes) do not. Similar mode mixing was also observed in the quasi-one-dimensional (quasi-1D) $K_{0.3}MoO_3$ using time-resolved pump-probe spectroscopy[59], and a simple model based on Ginzburg–Landau theory can well describe the entanglement of the electronic and lattice parts of the CDW order parameter. The similar phenomena observed in two systems with different

dimensionality suggest the importance of electron–phonon coupling in both materials.

However, a soft phonon is well established in $K_{0.3}MoO_3$[60], but shown to be absent in $CsV_3Sb_5$[20,21]. In the mean-field weak-coupling theory[28,41], the phonon softening, known as Kohn anomaly[61], is a direct consequence of the divergent electronic susceptibility, which screens the phonon vibration at the CDW wavevector. In reality, the singular electronic susceptibility is smeared out, especially at dimensions higher than one, and momentum dependent electron–phonon coupling dictates the phonon renormalization in certain systems such as $2H$-$NbSe_2$[62,63]. The lack of soft phonons in $CsV_3Sb_5$ seems to rule out both mechanisms. Instead, $CsV_3Sb_5$ may fall into the strong electron–phonon coupling regime[64], in which the non-detection of Kohn anomaly in the quasi-1D $(TaSe_4)_2I$, $NbSe_3$, and $BaVS_3$ has also been reported[65–67]. In all these materials, strong electron–phonon coupling tends to localize electrons, violating the adiabatic Born-Oppenheimer approximation[64] used in DFT. Failure of the conduction electrons to screen the phonon vibration can naturally explain the absence of phonon softening[68]. Possible phonon softening is also interrupted by the first-order nature of the CDW transition[53,69,70], precluding the observation of complete softening to zero frequency. The first-order transition may be understood as due to trilinear coupling of the three components ($3Q$) of the CDW[24] or the asymmetric double-well elastic potential for the ions[22], both contributing a term to the Landau free energy which is odd in the order parameter.

The strong-coupling nature of the CDW in $CsV_3Sb_5$ is indeed supported by multiple facts, according to the qualitative criteria discussed in ref. [28]. The CDW-induced gap $\Delta_{CDW}$ is large, with $2\Delta_{CDW}/k_BT_{CDW} \approx 22$ according to infrared spectroscopy[33], where $k_B$ is the Boltzmann constant. The lattice distortion is substantial (amounting to about 5% of the lattice constant[22]), the distorted lattice exhibits clustering of V atoms to form trimers and hexamers, and the CDW locks with the pristine lattice to form a commensurate structure, all indicating local chemical bonding[22,38]. Moreover, DFT shows that the elastic potential for the ions in the pristine structure features double minima deeper than the thermal energy $k_BT_{CDW}$ at the transition[22], a defining feature of the strong-coupling theory proposed by Gor'kov[64]. Such potential well traps the ions in one of its minima, precluding soft phonon condensation. From the perspective of Raman scattering, the electron–phonon coupling constant $\lambda$ can be estimated from the amplitude mode frequency $\omega_{AM}$ and the unscreened soft mode frequency $\omega_{SM}^0$ as $\lambda = (\omega_{AM}/\omega_{SM}^0)^2$, valid on the mean-field level[41]. The results for $CsV_3Sb_5$ and a variety of other CDW materials are compiled in Fig. 5. Notably, the four quasi-2D compounds $2H$-$NbSe_2$, $2H$-$TaSe_2$, $CsV_3Sb_5$, and $1T$-$TiSe_2$ are roughly located in the expected order according to their $T_{CDW}$, $\Delta_{CDW}$, and commensurability. The frequencies of the amplitude modes in $CsV_3Sb_5$ are large, only lower than that of the higher one in $1T$-$TiSe_2$. Although the exact value of $\lambda$ may not be meaningful beyond the weak-coupling limit, these results clearly indicate the strong-coupling nature of the CDW in $CsV_3Sb_5$.

Our Raman results offer informative insights into the CDW phase in $CsV_3Sb_5$, suggesting the dominance of the ISD-type distortion revealed by the CDW-induced modes and evidencing strong electron–phonon coupling. Although these results favor the local chemical bonding picture of the CDW transition, a coherent understanding of the mechanism, which should reconcile with the evidence for the electronically-driven scenario[22,30–37], is apparently called for. $CsV_3Sb_5$ represents a unique case in which the amplitude modes emerge in the

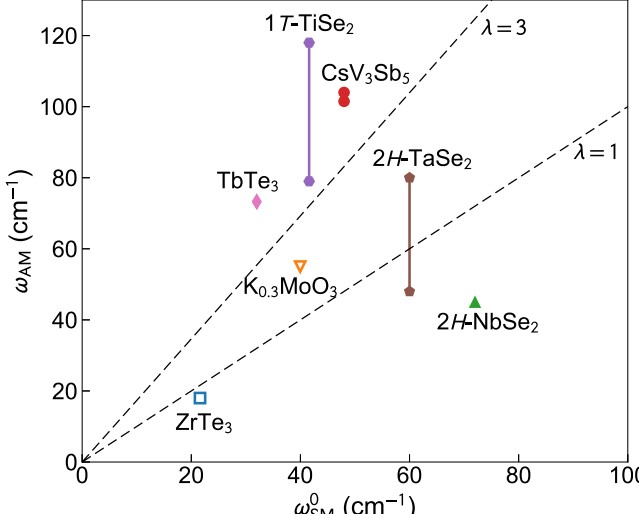

**Fig. 5 Evidence of strong-coupling CDW in $CsV_3Sb_5$.** Frequency of the amplitude mode $\omega_{AM}$ in the zero-temperature limit and the unscreened frequency of the soft mode $\omega_{SM}^0$ far above $T_{CDW}$ for a collection of CDW materials. Some of the materials feature two amplitude modes, hence two data points connected by a vertical line. Since no soft mode is observed in $CsV_3Sb_5$, $\omega_{SM}^0$ is taken to be its acoustic phonon frequency at 300 K[21]. Open (filled) symbols indicate the material is quasi-1D (quasi-2D). The dashed lines mark electron–phonon coupling constant $\lambda = 1$ and 3 according to mean-field theory. Source of data: $ZrTe_3$[80,81], $TbTe_3$[57,82], $K_{0.3}MoO_3$[43,60], $1T$-$TiSe_2$[47,83], $2H$-$TaSe_2$[45,46,84], $2H$-$NbSe_2$[84–86].

absence of soft phonons. As important collective excitations of the CDW ground state, how they form without being driven by folding of a soft phonon warrants further investigation. DFT accurately predicts the CDW-induced Raman modes in the zero-temperature limit, meanwhile showing the typical correlation of the amplitude modes and soft modes illustrated in Fig. 1a, suggesting that the experimentally elusive soft mode is somehow still relevant. Our work may stimulate further studies of the interplay between CDW amplitude modes and possible superconducting Higgs mode[48,49,71] in the kagome metals and the control of these symmetry-breaking states by ultrafast light[72–74].

## Methods

**Sample preparation.** $CsV_3Sb_5$ single crystals were synthesized using the flux method[11]. The freshly cleaved surface of the samples was used in the study of bulk crystals. Raman scattering spectroscopy was performed using home-built confocal microscopy setups in the back-scattering geometry with 532 nm laser excitation. The normally incident light was focused on the sample to a micron-sized spot, and the scattered light was directed through Bragg notch filters to access the low-wavenumber region. The Raman signal was collected using a grating spectrograph and a liquid-nitrogen-cooled charge-coupled device. The samples were mounted in a vacuum chamber during data acquisition. Temperature control was achieved using a Montana Instrument Cryostation.

**Calculations.** The DFT calculation results by Tan et al.[22] are used to compare with the experiment. In addition, we calculated the force constants by Vienna ab-initio Simulation Package (VASP)[75] and computed the phonon dispersion relation by Phonopy[76]. Perdew–Burke–Ernzerhof-type generalized gradient approximation (GGA) method has been used[77] and the projected augmented wave (PAW) potentials with 9 valence electrons for the Cs atom, 5 valence electrons for V and Sb atoms are employed. The DFT-D3 correction[78] is used to take interlayer van der Waals interactions into account. For the DFT calculation, a $5 \times 5 \times 5$ **k** mesh and an energy cutoff of 400 eV were used. For the pseudo-phonon spectra of the intermediate structures in Fig. 3, we used the same frozen phonon method to calculate force derivatives and obtained force constants, which is valid in the harmonic approximation. Specifically, for each intermediate structure we calculated the force differences between the slightly perturbed structure and the unperturbed one. The

representation decomposition relation in Eq. (1) is derived by directly calculating the characters of the folded modes (whose real space patterns are shown in Supplementary Fig. 7). See more details in Supplementary Note 3. The symmetry representations of Raman-active modes are calculated using the methods in ref. [79] as implemented in Bilbao Crystallographic Server.

## Data availability

The data in Figure 1e are provided in Supplementary Table 1. Other data are available from the corresponding authors upon reasonable request.

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

## Acknowledgements

This work was supported by the National Key Research and Development Program of China (Grant Nos. 2018YFA0307000 and 2017YFA0303201) and the National Natural Science Foundation of China (Grant Nos. 11774151). Growth of hexagonal boron nitride crystals was supported by the Elemental Strategy Initiative conducted by the MEXT, Japan (Grant No. JPMXP0112101001), JSPS KAKENHI (Grant No. JP20H00354), and A3 Foresight by JSPS. B.Y. acknowledges the financial support by the European Research Council (ERC Consolidator Grant "NonlinearTopo", No. 815869) and the ISF - Quantum Science and Technology (No. 1251/19).

## Author contributions

X.X. conceived the project. G.L., X.M., and K.H. performed the Raman experiments. Q.L., Y.D., and H.-H.W. grew the $CsV_3Sb_5$ crystals. K.W. and T.T. grew the h-BN crystals. G.L. and X.X. analyzed the experimental data. H.T., Y.L., and B.Y performed the DFT calculations. J.X., W.T., and L.G. performed atomic force microscopy measurements. X.X. and B.Y. interpreted the results and co-wrote the paper, with comments from all authors.

## Competing interests

The authors declare no competing interests.
