## [Peer Review File · Nature Communications]

REVIEWER COMMENTS

Reviewer #1 (Remarks to the Author):

The authors present a study of the kagome metal CsV₃Sb₅ using Raman scattering and first principles calculations. This is an extremely timely study, since the AV₃Sb₅ family of kagome metals are one of the most exciting topics of the past couple of years. The measurements are carefully performed, and the first principles calculations seem to be of high quality. Nevertheless, I do not suggest this paper for publication in Nature Communications, at least in its current form, for the reasons detailed below:

i) There are some points in the discussion of the charge density wave transitions that I find confusing. Perhaps the most important one is the discussion of a soft acoustic mode. The authors are surely aware, as they discuss elsewhere in the text, that the phonons responsible of the transition (or can at least be associated with them) are zone boundary phonons. Is it meaningful to talk about acoustic phonons at the zone boundary? My understanding is that acoustic modes are only well-defined near the zone center. If the authors disagree with my statement, then they should clarify how they define an acoustic zone-boundary phonon.

ii) A point that is raised a few times is that the CDW transition is not accompanied by the condensation of a 'soft' mode in this compound. The definition of the word 'soft' is often a bit lax in the field. What do the authors exactly mean by it: Do they mean that there is no low frequency-phonon, or just a phonon the frequency of which goes to zero at the transition temperature? While the first one clearly doesn't exist, I fail to see why this is an important point that is worth mentioning in the abstract of the paper: The transition in this compound is shown to be first order, which by its very definition, does not require a phonon frequency going to zero.

iii) The discussion in the beginning of page 6 is misleading for several reasons. The authors state that "Fig. 3a shows that after the $2 \times 2 \times 1$

CDW transition, (the M₁₊ modes) are folded to Γ and form triply degenerate modes." This does not happen after the transition, instead, this happens when the authors consider a supercell without a structural distortion; so use a smaller Brillouin zone to represent the phonons of the high-symmetry & high-temperature structure. Such a 3-fold degeneracy at the zone center cannot really exist in this system ever, since hexagonal space groups don't allow 3-fold degeneracies at the zone center.

iv) (Continuing from the previous point) Eq. 1 is not completely correct, since on the left hand side they have a space group representation, whereas on the right hand side they have point group irreps. But it is nevertheless understandable since it is common in literature to refer to zero-wavevector phonons in solids by the point group labels. What is missing, however, is how Eq. 1 is obtained. Is it obvious that the $M1+$ folds onto these modes? There are formal ways to achieve expressions like this; but the authors don't explain what calculation they performed.

v) The manuscript uses the phrase "phonon wavefunction" to referring to figures 4, etc. This is a rather nonstandard usage - in the Born-von Karman approach, which the authors certainly use, the positions of the atoms are treated as classical parameters, and phonons are technically not quantum objects. Referring to a dynamical matrix eigenvector, or a phonon displacement pattern, would be more correct.

vi) On Fig. 3, results of phonon calculations for intermediate states are presented. These are not crystal structures which are relaxed, so the forces on the atoms are not zero. The authors need to explain what exactly they mean by phonons in these structures, and what calculating them involved.

vii) The calculation methods are not explained in detail, instead, the authors refer to Ref. 22 for details. However, there are many calculations that are not included in Ref. 22 (or its supplementary information). I think this paper deserves a more detailed methods section.

viii) An important shortcoming of the paper is that it does not see any signature of a second phase transition below the 94 K one. There seems to be an increasing amount of new results that support this transition, and this referee is inclined to believe that there are indeed two transitions. The present manuscript mentions this point, but without citing all the experimental evidence for the second transition, or providing any explanation (or even speculation) on the reason why they don't see it. Is the samples used in this study different from those used in these other studies? Or, is there really a consensus on the presence of two transitions? What may the reason for this disagreement be?

ix) There is an editorial question about impact and suitability of this paper. Nature Communications has an impact factor of about 15, and this paper is surely to collect that many citations per year because of how quickly this field is developing. If this is the necessary condition for being suitable for Nature Communications, then there is no question. However, I don't think this is a manuscript suitable for a general interest readership: As much as the data and the information in this paper is likely to be of interest to other people working on this compound, I don't think a general readership - even a general condensed matter readership - would find this paper interesting for themselves. As a result, I suggest this paper to be transferred to a more specialized journal.

Reviewer #2 (Remarks to the Author):

In the manuscript entitled “Observation of anomalous amplitude modes in the kagome metal CsV₃Sb₅” by Liu et al., the authors use Raman spectroscopy to study the charge density wave transition in the newly-discovered kagome metal system CsV₃Sb₅. Below the critical temperature of the transition, they find new Raman-active modes appear and associate them with CDW amplitude modes. They also find strong hybridization between the amplitude modes and other phonon modes in the crystal, and account for this through strong electron-phonon coupling.

Understanding the specific details of the CDW order in the new AV₃Sb₅ material family will be crucial for understanding the more exotic phases that appear at lower temperatures, like superconductivity and purported time-reversal symmetry breaking order. This manuscript adds important new details to the discussion. For that reason, I support publication in Nature Communications if the authors can satisfactorily address the following comments:

(1) The authors have said very little about the c-axis modulation of the CDW order, which is now completely established experimentally (although the exact nature of the c-axis ordering is still up for debate). They should discuss the specific implications of c-axis CDW modulation on their Raman spectroscopy data. In particular, they focus on folding of M-point phonons to Gamma, but what about L-point phonons? Those phonon frequencies should also appear below T_{CDW} .

(2) The authors repeatedly highlight the absence of acoustic phonon softening, but this is in fact not surprising. Due to the 3Q-CDW nature in the AV₃Sb₅ materials, there is a tri-linear coupling of Q₁, Q₂ and Q₃ CDW amplitudes in the Landau free energy. This cubic term directly leads to a first-order phase transition (rather than a second-order phase transition in conventional CDW systems). At a first order phase transition, the order parameters “jump” from 0 to a finite value, and there is no phonon softening associated with this transition. The authors should address this in the context of whether or not acoustic phonon softening is actually relevant in this system.

(3) Ever since the AV₃Sb₅ discovery paper, it appears that the community has been “locked” into calling the phase transition a CDW transition. However, it is becoming increasingly clear (and this manuscript supports the idea) that there is strong electron-phonon coupling in this material, and the transition could equally well be described as a structural phase transition. It is a bit of a chicken-or-egg question, but the authors should emphasize whether or not their data can distinguish a CDW transition from a structural transition, or if it is even useful to make such a distinction.

Reviewer #3 (Remarks to the Author):

Gian Liu et al. proposes a Raman study supported by DFT calculations to identify and explain the behavior of collective charge density modes in the compound CsV₃Sb₅.

The originality of this study consists in saying that CDW collective modes are detectable by Raman scattering without the associated soft modes usually observable by X-ray or neutron experiments.

The reasons of this experimental fact are not very clear to me.

I would like the authors to be more explicit:

Is this due to the strong electron phonon coupling?

I would also like the authors to clarify what they mean by hybridization between amplitude modes and zone folded modes. From what I know, the amplitude mode is a folded mode induced by the soft acoustic mode which undergoes the constraint of the gap at the zone-center.

The authors claim that their data shows that the A_{1g} mode survives beyond 60 K,

(see, end of page 6) if I look at figure 2c this does not seem to be the case.

The frequency dependences of A₂ and E₃ modes called by the authors CDW amplitude modes seem very similar to those of E₁-E₄ and A₃-A₅. Why the authors focus on these in particular?

The Authors should also mention the following references on collective modes in dichalcogenides observed by Raman:

Phys. Rev. Lett. 122, 127001 – Published 27 March 2019

Phys. Rev. B 89, 060503(R) – Published 18 February 2014

I am waiting for clarification on these points mentioned above, before deciding whether or not I support this work.

regards

Response to Reviewer 1

The authors present a study of the kagome metal CsV_3Sb_5 using Raman scattering and first principles calculations. This is an extremely timely study, since the AV_3Sb_5 family of kagome metals are one of the most exciting topics of the past couple of years. The measurements are carefully performed, and the first principles calculations seem to be of high quality. Nevertheless, I do not suggest this paper for publication in Nature Communications, at least in its current form, for the reasons detailed below:

We appreciate the reviewer’s effort to evaluate our work and are grateful for the constructive comments. With our response below and revisions of the manuscript, we have clarified several issues raised by the reviewer. We hope that, with these improvements, the reviewer could have a different view on our work.

i) There are some points in the discussion of the charge density wave transitions that I find confusing. Perhaps the most important one is the discussion of a soft acoustic mode. The authors are surely aware, as they discuss elsewhere in the text, that the phonons responsible of the transition (or can at least be associated with them) are zone boundary phonons. Is it meaningful to talk about acoustic phonons at the zone boundary? My understanding is that acoustic modes are only well-defined near the zone center. If the authors disagree with my statement, then they should clarify how to they define an acoustic zone-boundary phonon.

We agree with the reviewer that acoustic phonons are only well-defined near the zone center, but not at the zone-boundary. What we actually meant was that softening may occur in a branch that can be defined as acoustic near the zone center.

Since the emphasis on “acoustic” is not critical, to avoid confusion, we have removed “acoustic” and simplified the term as “soft mode” in most places in the manuscript. Details of the revision are highlighted in the revised version.

ii) A point that is raised a few times is that the CDW transition is not accompanied by the condensation of a ‘soft’ mode in this compound. The definition of the word ‘soft’ is often a bit lax in the field. What do the authors exactly mean by it: Do they mean that there is no low frequency phonon, or just a phonon the frequency of which goes to zero at the transition temperature? While the first one clearly doesn’t exist, I fail to see why this is an important point that is worth mentioning in the abstract of the paper: The transition in this compound is shown to be first order, which by its very definition, does not require a phonon frequency going to zero.

The meaning of ‘soft’ mode can indeed be defined more clearly. To remove ambiguity, we state in the introduction of the revised manuscript that soft modes are those “showing frequency softening upon cooling towards a phase transition”.

The absence of soft phonons in AV_3Cs_5 was first reported in Ref 20 [Phys. Rev. X 11, 031050 (2021)] and stressed to be a feature of the unconventional CDW in this material family. Indeed, this is unconventional in the sense that CDW transitions in many materials are triggered by soft modes whose frequency go to zero at T_{CDW} . Understanding the role of soft modes, or the cause for their absence, may hold the key to unraveling the CDW mechanism.

We agree with the reviewer that the first-order nature of the transition can make phonon softening difficult to observe. This view is also shared by the second reviewer. However, we believe the discussion of the relevance of phonon softening is still not trivial. First-order transition does not explain why the experiments in Refs 20 and 21 actually do not find appreciable phonon softening between 300 K and 50 K. Although complete softening to zero frequency should be interrupted by a first-order transition, one would expect frequency softening (not necessarily to zero) to be still present. We think one possible explanation for the absence of clear phonon softening is due to strong electron-phonon coupling. As L.P. Gor'kov pointed out in his work (Ref. 55 of the previous version), strong enough electron-phonon coupling can lead to breakdown of the adiabatic approximation, such that conduction electrons cannot screen the phonon vibration at the CDW wavevector.

As the reviewer suggested, we have revised the abstract by reducing the emphasis on phonon softening absence. We have also added a few sentences about the first-order nature of the transition in the Discussion section, “Possible phonon softening is also interrupted by the first-order nature of the CDW transition [66–68], precluding the observation of complete softening to zero frequency. The first-order transition may be understood as due to trilinear coupling of the three components ($3Q$) of the CDW [24] or the asymmetric double-well elastic potential for the ions [22], both contributing a term to the Landau free energy which is odd in the order parameter.”

iii) The discussion in the beginning of page 6 is misleading for several reasons. The authors state that “Fig. 3a shows that after the $2 \times 2 \times 1$ CDW transition, (the M_1^+ modes) are folded to Γ and form triply degenerate modes.” This does not happen after the transition, instead, this happens when the authors consider a supercell without a structural distortion; so use a smaller Brillouin zone to represent the phonons of the high-symmetry & high-temperature structure. Such a 3-fold degeneracy at the zone center cannot really exist in this system ever, since hexagonal space groups don't allow 3-fold degeneracies at the zone center.

We agree with the reviewer for this comment. Before CDW distortion, the triple degenerate states does not exist in reality, because they are due to artificial band folding. Only when CDW appears, they come out, characterizing the CDW transition as amplitude modes. To avoid confusion we change the sentence “Fig. 3a shows that they form triply

degenerate modes at Γ due to the artificial band folding without lattice distortion, in which these modes are not measurable in Raman. Only when CDW appears, they come out as amplitude modes, characterizing the CDW transition.”.

iv) (Continuing from the previous point) Eq. 1 is not completely correct, since on the left hand side they have a space group representation, whereas on the right hand side they have point group irreps. But it is nevertheless understandable since it is common in literature to refer to zero-wavevector phonons in solids by the point group labels. What is missing, however, is how Eq. 1 is obtained. Is it obvious that the M_1^+ folds onto these modes? There are formal ways to achieve expressions like this; but the authors don't explain what calculation they performed.

We follow the reviewer's suggestion to add more detailed discussions on this point, now included in the revised Supplementary Information. We first calculated the character table of the three folded modes $3M_1^+$ (whose real space patterns are shown in Supplementary Figure 7) as shown in Tab. I. The representation of $3M_1^+$ is reducible and its decomposition onto each irreducible representation (irreps) can be done by standard textbook group theory method:

$$3M_1^+ = \sum_{\oplus} a_i \Gamma_i,$$

$$a_i = \frac{1}{24} \sum_{g \in D_{6h}} \chi_{\Gamma_i}^*(g) \chi_{3M_1^+}(g),$$

where $\chi_{\Gamma_i}(g)$ refers to the character of the symmetry operator g belonging to irreps Γ_i . Combining the characters of $\chi_{3M_1^+}(g)$ shown in Tab. I and the irreps table of D_{6h} , we found that the only nonzero coefficients are: $a_{A_{1g}} = a_{E_{2g}} = 1$.

	E	$2C_6$	$2C_3$	C_2	$3C_2'$	$3C_2''$	i	$2S_3$	$2S_6$	σ_h	$3\sigma_d$	$3\sigma_v$
$3M_1^+$	3	0	0	3	1	1	3	0	0	3	1	1

TABLE I. Character table of $3M_1^+$ under D_{6h} .

v) The manuscript uses the phrase “phonon wavefunction” to referring to figures 4, etc. This is a rather nonstandard usage - in the Born-von Karman approach, which the authors certainly use, the positions of the atoms are treated as classical parameters, and phonons are technically not quantum objects. Referring to a dynamical matrix eigenvector, or a phonon displacement pattern, would be more correct.

The reviewer makes a good point. We have changed this phrase to “phonon displacement pattern” when discussing the atomic vibrations and to “dynamical matrix eigenvector” or

simply “eigenvector” when discussing the mode hybridization.

vi) On Fig. 3, results of phonon calculations for intermediate states are presented. These are not crystal structures which are relaxed, so the forces on the atoms are not zero. The authors need to explain what exactly they mean by phonons in these structures, and what calculating them involved.

The phonon band structures in Fig. 3 are directly calculated by DFT based on the intermediate crystal structures between the nonrelaxed (0%) and the fully relaxed (100%), so the meaning of phonons is the same as that in the nonrelaxed case, with imaginary modes indicating the instability of the structure. We agree with the reviewer that for those intermediate structures the forces on the atoms are nonzero and this is the reason that the intermediate structure still has apparent imaginary modes (*e.g.* 0%, 10% and 40%). In the revised manuscript, we add this information to the figure caption.

vii) The calculation methods are not explained in detail, instead, the authors refer to Ref. 22 for details. However, there are many calculations that are not included in Ref. 22 (or its supplementary information). I think this paper deserves a more detailed methods section.

We thank the reviewer for the suggestion. We have added more details in the calculation methods and in the Supplementary Information. Please see details in the revised version.

viii) An important shortcoming of the paper is that it does not see any signature of a second phase transition below the 94 K one. There seems to be an increasing amount of new results that support this transition, and this referee is inclined to believe that there are indeed two transitions. The present manuscript mentions this point, but without citing all the experimental evidence for the second transition, or providing any explanation (or even speculation) on the reason why they don’t see it. Is the samples used in this study different from those used in these other studies? Or, is there really a consensus on the presence of two transitions? What may the reason for this disagreement be?

We actually briefly discussed the unidirectional order in the paragraph before the Discussion section. The main experimental evidence for this order was also cited there, *i.e.* Refs 19, 25, and 26 in the previous version. Ref 27 also reported the observation of this order, and in the revised manuscript we cite it with the other three references.

All these independent studies reported the unidirectional order based on scanning tunneling microscopy, and we think this can be regarded as consensus in the field on its presence and the second transition that leads to it. However, Raman spectroscopy may not be sensitive enough to detect this order. This may be due to the surface origin of the unidirectional order, as proposed in Ref 26 and in arXiv:2109.03418. As the CDW-induced modes related with the 2×2 order are already weaker than those observed in some other well-known CDW

systems (such as the transition metal dichalcogenides, see e.g. Refs 39–44 in the previous version), the surface modes could be even more challenging to pick up by Raman spectroscopy. Ref 26 also reported that the unidirectional order exists as nanoscale domains, so this does not seem like long-range order. In this case it may be meaningless to discuss “modes” that can be detected by Raman spectroscopy, which is a macroscopic probe of collective excitation.

To clarify this point, we add the following sentence to the paragraph preceding the Discussion section, “Raman scattering, as a bulk probe, is probably not sensitive enough to the unidirectional order, due to its possible surface origin [26, arXiv:2109.03418] and its existence in nanoscale domains [26].”

ix) There is an editorial question about impact and suitability of this paper. Nature Communications has an impact factor of about 15, and this paper is surely to collect that many citations per year because of how quickly this field is developing. If this is the necessary condition for being suitable for Nature Communications, then there is no question. However, I don’t think this is a manuscript suitable for a general interest readership: As much as the data and the information in this paper is likely to be of interest to other people working on this compound, I don’t think a general readership - even a general condensed matter readership - would find this paper interesting for themselves. As a result, I suggest this paper to be transferred to a more specialized journal.

We stress that the general interest of our work lies in the following areas.

It reveals an important aspect of the possible mechanisms for the CDW transition in a material of general interest to the condensed matter physics and materials science communities. So far the majority of reported works take the view that the CDW in this material family is driven by Fermi surface instability (see Refs 22 and 30–37 in the revised manuscript). We show here clear experimental evidence of strong electron-phonon coupling, emphasizing the important role of the lattice degree of freedom. We believe such evidence and the discussions on the CDW mechanism could be of general interest to a wide readership.

It establishes the existence of collective excitations in this intriguing material family, which is the starting point to uncover more novel phenomena and to harness light control of the ground states. An example of the former is the observation of superconducting Higgs modes due to the coupling to the CDW amplitude modes [see e.g. Phys. Rev. Lett. 122, 127001 (2019) and references therein, as recommended by the third reviewer]. The study of Higgs mode in condensed matter systems is gaining much interest recently [see an review in Annu. Rev. Condens. Matter Phys. 11, 103 (2020)], and the kagome superconductors provide a new exciting platform.

Light control of quantum materials is also another trending topic in condensed matter and optical physics, and CDW materials are popular platforms for its realization. In such

systems, the photo-induced non-equilibrium phenomena typically involve amplitude modes [see examples in PNAS 117, 8788 (2020) and Science 321, 1649 (2008), and a review in J. Phys.: Condens. Matter 33, 353001 (2021)]. Given the current interest in the kagome metals, we believe our work will stimulate further exploration along this direction.

These are a few examples that we see the potential impact of our work, which we believe makes it of general interest to a wide readership in condensed matter physics, materials science, and optical science. We therefore believe it is suitable for Nature Communications. We have added the following sentence in the finishing paragraph in the revised manuscript, “Our work may stimulate further studies of the interplay between CDW amplitude modes and possible superconducting Higgs mode [48, 49, 69] in the kagome metals and the control of these symmetry-breaking states by ultrafast light [70–72].”

Response to Reviewer 2

In the manuscript entitled “Observation of anomalous amplitude modes in the kagome metal CsV_3Sb_5 ” by Liu et al., the authors use Raman spectroscopy to study the charge density wave transition in the newly-discovered kagome metal system CsV_3Sb_5 . Below the critical temperature of the transition, they find new Raman-active modes appear and associate them with CDW amplitude modes. They also find strong hybridization between the amplitude modes and other phonon modes in the crystal, and account for this through strong electron-phonon coupling.

Understanding the specific details of the CDW order in the new AV_3Sb_5 material family will be crucial for understanding the more exotic phases that appear at lower temperatures, like superconductivity and purported time-reversal symmetry breaking order. This manuscript adds important new details to the discussion. For that reason, I support publication in Nature Communications if the authors can satisfactorily address the following comments:

We thank the reviewer for taking time to read our manuscript and for pointing out the potential interest of our work to the community. In the following we address the specific questions.

(1) The authors have said very little about the c -axis modulation of the CDW order, which is now completely established experimentally (although the exact nature of the c -axis ordering is still up for debate). They should discuss the specific implications of c -axis CDW modulation on their Raman spectroscopy data. In particular, they focus on folding of M -point phonons to Γ , but what about L -point phonons? Those phonon frequencies should also appear below T_{CDW} .

We totally agree with the reviewer on this point. The c -axis modulation should in principle lead to new Raman modes. However, because most of the modes (except for the lowest frequency A_1 mode) observed in our experiment are highly consistent with the DFT results for a single-layer model, we conclude that the Raman response is dominated by intralayer rather than interlayer contributions. The almost non-detection of modes folded from the L -point may be attributed to weak interlayer interaction, because the M - and L -point instabilities differ only in the interlayer ordering.

These $2 \times 2 \times 2$ and $2 \times 2 \times 4$ phases discussed in literature are expected to exhibit much more Raman-active modes than the $2 \times 2 \times 1$ phase. However, our experiment does not reveal those additional modes. It indicates that those c -axis related modes may exist but exhibit very weak signals in Raman.

Below we consider Raman-active modes for different forms of c -axis modulation to further support this point. At Γ point all phonon modes can be classified into acoustic and optic

modes, *i.e.* $\Gamma_{\text{total}} = \Gamma_{\text{acoustic}} \oplus \Gamma_{\text{optic}}$. The optic phonon modes can be further classified into infra-red (IR) active, Raman active, and silent modes as $\Gamma_{\text{optic}} = \Gamma_{\text{IR}} \oplus \Gamma_{\text{Raman}} \oplus \Gamma_{\text{silent}}$. Based on group theory analysis [Kroumova *et al.* *Phase Transitions*, 76, 155 (2003)], we have derived the symmetry representations of phonon modes for five different modulations along the c axis:

- (i) $2 \times 2 \times 1$ CDW composed of inverse star of david (ISD) structures with no modulation along the c axis (Tab. II),
- (ii) $2 \times 2 \times 2$ CDW composed of ISD structures having an interlayer π phase shift (Tab. III),
- (iii) $2 \times 2 \times 2$ CDW composed of ISD and SD structures having no phase shift (Tab. IV),
- (iv) $2 \times 2 \times 2$ CDW composed of ISD and SD structures having π -phase shift (Tab. V),
- (v) $2 \times 2 \times 4$ CDW composed of one ISD and three consecutive layers of SD structures having no phase shift (Tab. VI).

These structures either have the D_{6h} or D_{2h} point groups. For the former (latter), the Raman-active modes that can be observed in the back-scattering geometry of our experiment are the A_{1g} and E_{2g} (A_g and B_{1g}) modes. Except for the $2 \times 2 \times 1$ structure, the predicted number of modes far exceeds that observed experimentally. Therefore, in terms of Raman response, the c -axis modulation is not clearly manifested.

	A_{1g}	A_{2g}	B_{1g}	B_{2g}	E_{1g}	E_{2g}	A_{1u}	A_{2u}	B_{1u}	B_{2u}	E_{1u}	E_{2u}
Γ_{total}	5	3	4	2	6	8	1	9	5	7	14	8
Γ_{acoustic}								1			1	
Γ_{optic}	5	3	4	2	6	8	1	8	5	7	13	8
Γ_{IR}								8			13	
Γ_{Raman}	5				6	8						
Γ_{silent}		3	4	2			1		5	7		8

TABLE II. Symmetry table of phonon modes of $2 \times 2 \times 1$ CDW with ISD structures. The point group is D_{6h} .

	A_g	B_{1g}	B_{2g}	B_{3g}	A_u	B_{1u}	B_{2u}	B_{3u}
Γ_{total}	14	12	12	10	8	16	17	19
Γ_{acoustic}						1	1	1
Γ_{optic}	14	12	12	10	8	15	16	18
Γ_{IR}						15	16	18
Γ_{Raman}	14	12	12	10				
Γ_{silent}					8			

TABLE III. Symmetry table of phonon modes of $2 \times 2 \times 2$ CDW with ISD structures but having interlayer π -phase shift. The point group is D_{2h} .

In principle, the D_{6h} and D_{2h} point groups can be distinguished by polarization angle dependent measurements. We consider the collinear polarization configuration (denoted as

	A_{1g}	A_{2g}	B_{1g}	B_{2g}	E_{1g}	E_{2g}	A_{1u}	A_{2u}	B_{1u}	B_{2u}	E_{1u}	E_{2u}
Γ_{total}	12	6	9	5	15	17	2	16	9	13	25	15
Γ_{acoustic}								1			1	
Γ_{optic}	12	6	9	5	15	17	2	15	9	13	24	15
Γ_{IR}								15			24	
Γ_{Raman}	12				15	17						
Γ_{silent}	6	9	5			2		9	13		15	

TABLE IV. Symmetry table of phonon modes of $2 \times 2 \times 2$ CDW with ISD and SD structures having no interlayer shift. The point group is D_{6h} .

	A_g	B_{1g}	B_{2g}	B_{3g}	A_u	B_{1u}	B_{2u}	B_{3u}
Γ_{total}	29	23	24	20	17	31	34	38
Γ_{acoustic}						1	1	1
Γ_{optic}	29	23	24	20	17	30	33	37
Γ_{IR}						30	33	37
Γ_{Raman}	29	23	24	20				
Γ_{silent}					17			

TABLE V. Symmetry table of phonon modes of $2 \times 2 \times 2$ CDW with ISD and SD structures having π -phase interlayer shift. The point group is D_{2h} .

XX), in which the polarizations for the incident and scattered light are kept parallel while they are co-rotated with respect to a given crystal axis. For the D_{6h} point group, the A_{1g} and E_{2g} Raman-active modes that contribute to the back-scattering response have the following Raman tensors,

$$\mathbf{R}_{A_{1g}} = \begin{pmatrix} a & 0 & 0 \\ 0 & a & 0 \\ 0 & 0 & b \end{pmatrix}; \quad \mathbf{R}_{E_{2g}} = \begin{pmatrix} 0 & f & 0 \\ f & 0 & 0 \\ 0 & 0 & 0 \end{pmatrix}, \begin{pmatrix} f & 0 & 0 \\ 0 & -f & 0 \\ 0 & 0 & 0 \end{pmatrix}. \quad (1)$$

The Raman scattering intensity of both modes are expected to be independent of the angle of the linear polarization θ , $I_{A_{1g}}(\theta) \propto a^2$, $I_{E_{2g}}(\theta) \propto f^2$. For the D_{2h} point group, the relevant modes have A_g and B_{1g} symmetries, and their Raman tensors are

$$\mathbf{R}_{A_g} = \begin{pmatrix} a & 0 & 0 \\ 0 & b & 0 \\ 0 & 0 & c \end{pmatrix}; \quad \mathbf{R}_{B_{1g}} = \begin{pmatrix} 0 & d & 0 \\ d & 0 & 0 \\ 0 & 0 & 0 \end{pmatrix}. \quad (2)$$

The Raman scattering intensity of both modes are expected to be anisotropic, $I_{A_g}(\theta) \propto (a \cos^2 \theta + b \sin^2 \theta)^2$, $I_{B_{1g}}(\theta) \propto d^2[1 - \cos(4\theta)]/2$.

Fig. R1 shows the polarization-angle dependent data of CsV_3Sb_5 in the CDW phase (at 4 K) and the normal phase (at 100 K). We focus on the main lattice modes due to

	A_{1g}	A_{2g}	B_{1g}	B_{2g}	E_{1g}	E_{2g}	A_{1u}	A_{2u}	B_{1u}	B_{2u}	E_{1u}	E_{2u}
Γ_{total}	26	10	20	12	35	33	6	30	16	24	45	31
Γ_{acoustic}								1			1	
Γ_{optic}	26	10	20	12	35	33	6	29	16	24	44	31
Γ_{IR}								29			44	
Γ_{Raman}	26				35	33						
Γ_{silent}		10	20	12			6		16	24		31

TABLE VI. Symmetry table of phonon modes of $2 \times 2 \times 4$ CDW with one ISD and three consecutive layer of SD structures having no interlayer phase shift. The point group is D_{6h} .

FIG. R1. (a) and (b) Polarization angle dependent Raman intensity color plots for the CDW and normal phases, taken at 4 K and 100 K, respectively, for the collinear polarization configuration. (c) and (d) Polarization angle dependence of the intensity of the main lattice modes. The symbols are analyzed values from Lorentzian peak fitting. The circles represent average values of the data points. Small variations are present due to experimental uncertainties. The standard deviation of the analyzed angle-dependent intensity divided by the average value yields the following results: 4.0% (E_{2g} , 4 K), 5.5% (E_{2g} , 100 K), 4.3% (A_{1g} , 4 K), and 5.3% (A_{1g} , 100 K). The slightly lower values at 4 K can be attributed to enhanced signal-to-noise ratio due to the strengthening of the main lattice phonon peaks, which do not support CDW-induced D_{2h} point group.

their excellent signal-to-noise ratio. Neither modes show appreciable polarization angle dependence, above and below the CDW transition. We therefore conclude that either the c -axis modulation is too weak to induce clear polarization angle dependence on the observed Raman modes, or those candidate stacking orders with the D_{2h} point group (ii and iv) can be ruled out, because the B_{1g} mode intensity should vary sharply between zero and the maximum value, which is incompatible with the observed results.

We have included these discussions in the revised manuscript and the Supplementary Information.

(2) The authors repeatedly highlight the absence of acoustic phonon softening, but this is in fact not surprising. Due to the $3Q$ -CDW nature in the AV_3Sb_5 materials, there is a tri-linear coupling of Q_1 , Q_2 and Q_3 CDW amplitudes in the Landau free energy. This cubic term directly leads to a first-order phase transition (rather than a second-order phase transition in conventional CDW systems). At a first order phase transition, the order parameters “jump” from 0 to a finite value, and there is no phonon softening associated with this transition. The authors should address this in the context of whether or not acoustic phonon softening is actually relevant in this system.

We thank the reviewer for pointing out this important view, which was also shared by the first reviewer. We understand that the first-order nature of the transition is now an established fact. But we do not think it makes the discussion of phonon softening (or its absence) trivial. The trilinear coupling discusses the phase transition by approaching T_{CDW} from below. We discuss the transition when approaching it from above. First-order transition does not explain why the experiments in Refs 20 and 21 actually do not find appreciable phonon softening between 300 K and 50 K. Although complete softening to zero frequency should be interrupted by a first-order transition, one would expect frequency softening (not necessarily to zero) to be still present. We think one possible explanation for the absence of clear phonon softening is due to strong electron-phonon coupling. As L.P. Gor'kov pointed out in his work (Ref. 55 in the previous version), strong enough electron-phonon coupling can lead to breakdown of the adiabatic approximation, such that conduction electrons cannot screen the phonon vibration at the CDW wavevector.

We think phonon softening is still relevant in this system, although the softening was not clearly observed experimentally. The relevance is more apparent in the DFT calculations, which clearly shows the typically expected connection between the soft mode and amplitude modes.

We agree with the reviewer that the trilinear term coupling the $3Q$ order parameters can make the transition first order, as discussed in detail in Ref 24. We think the occurrence of a first-order transition can also be understood from another perspective, namely, the ionic potential profile as a function of lattice distortion exhibits an asymmetric double-well shape, indicating the presence of an odd-order term in the Landau free energy expansion. Microscopically, the cause of this imbalance between the SD and ISD can be attributed to the specific electron-phonon coupling in this system.

To reflect these points, we have made the following changes to the manuscript.

(i) We have revised the abstract by reducing emphasis on the absence of phonon softening, but stressing the strong electron-phonon coupling revealed here.

(ii) We have added a few sentences about the first-order nature of the transition in the Discussion section, “Possible phonon softening is also interrupted by the first-order nature of the CDW transition [66–68], precluding the observation of complete softening to zero

frequency. The first-order transition may be understood as due to trilinear coupling of the three components ($3Q$) of the CDW [24] or the asymmetric double-well elastic potential for the ions [22], both contributing a term to the Landau free energy which is odd in the order parameter.”

(iii) In the Discussion section, we also add “DFT accurately predicts the CDW-induced Raman modes in the zero-temperature limit, meanwhile showing the typical correlation of the amplitude modes and soft modes illustrated in Fig. 1a, suggesting that the experimentally elusive soft mode is somehow still relevant.”

(3) Ever since the AV_3Sb_5 discovery paper, it appears that the community has been “locked” into calling the phase transition a CDW transition. However, it is becoming increasingly clear (and this manuscript supports the idea) that there is strong electron-phonon coupling in this material, and the transition could equally well be described as a structural phase transition. It is a bit of a chicken-or-egg question, but the authors should emphasize whether or not their data can distinguish a CDW transition from a structural transition, or if it is even useful to make such a distinction.

The reviewer probably would like to discuss whether the phase transition is driven by the electronic or the lattice degree of freedom. This is a very interesting and difficult question. What we observed in Raman scattering are phonons, so our results clearly show the relevance of the lattice degree of freedom. Moreover, the observed large amplitude mode frequencies and mode mixing in our work, together with various other features of the system (see details in the Discussion section), point to strong electron-phonon coupling. This scenario emphasizes the dominance of local chemical bonding in driving the transition, or in the language of the reviewer, it is more like a structural phase transition. However, abundant evidence also exists which supports the electronically-driven picture (see e.g. optical and ARPES studies in Refs 30–34 in the previous version). How to reconcile these two different views is still a question to be answered in the future.

Limited by the qualitative understanding of the electron-phonon coupling in this system, we would like to refrain from over-interpreting our results and making too strong statements. Instead, we briefly point out this important question in the Discussion section, “Although these (Raman) results favour the local chemical bonding picture of the CDW transition, a coherent understanding of the mechanism, which should reconcile with the evidence for the electronically-driven scenario, is apparently called for.”

Response to Reviewer 3

Gan Liu et al. proposes a Raman study supported by DFT calculations to identify and explain the behavior of collective charge density modes in the compound CsV_3Sb_5 . The originality of this study consists in saying that CDW collective modes are detectable by Raman scattering without the associated soft modes usually observable by X-ray or neutron experiments. The reasons of this experimental fact are not very clear to me. I would like the authors to be more explicit: Is this due to the strong electron phonon coupling?

Yes, that is what we try to convey in our manuscript. The amplitude modes are identified according to their temperature dependence and by following how the DFT-calculated soft mode evolves as the lattice distortion is increased. However, the predicted soft mode actually has not been detected experimentally (Refs 20 and 21). We believe this is due to strong electron-phonon coupling, as proposed by L.P. Gor'kov in Ref 55 in the previous version. The key point is that strong enough electron-phonon coupling can lead to breakdown of the adiabatic approximation used in DFT, such that conduction electrons can no longer screen the phonon vibration at the CDW wavevector. We discuss this possibility in the second paragraph in the Discussion section. Another possible cause for the absence of a soft mode is due to the first-order nature of the transition, as the other two reviewers pointed out. We also include this possibility in the revised Discussion section.

How to understand the emergence of amplitude modes in CsV_3Sb_5 is still an open question. In the concluding paragraph, we revise the text to reflect this point, “ CsV_3Sb_5 represents a unique case in which the amplitude modes emerge in the absence of soft phonons. As important collective excitations of the CDW ground state, how they form without being driven by folding of a soft phonon warrants further investigation. DFT accurately predicts the CDW-induced Raman modes in the zero-temperature limit, meanwhile showing the typical correlation of the amplitude modes and soft modes illustrated in Fig. 1a, suggesting that the experimentally elusive soft mode is somehow still relevant. ”

I would also like the authors to clarify what they mean by hybridization between amplitude modes and zone folded modes. From what I know, the amplitude mode is a folded mode induced by the soft acoustic mode which undergoes the constraint of the gap at the zone-center.

We agree with the reviewer on the understanding of the definition of the amplitude modes. As for the zone-folded modes, we refer to those modes folded to the zone center from the optical branches by the superlattice formation, namely, those modes always at positive frequencies at the Γ point in Fig. 3a of the manuscript. These definitions have been discussed in the section “Nature of CDW-induced modes”.

What we mean by hybridization is that vibration patterns for the amplitude modes

FIG. R2. Raman intensity color plots taken from the main text (left) and from the Supplementary Information (right). The green ovals highlight the overdamped amplitude mode, which extends beyond 60 K. This mode is absent in the 8.3 nm sample, because the lower sample quality suppresses the CDW.

and zone-folded modes share similar features, as demonstrated by the calculated phonon displacement patterns shown in Fig. 4. The hybridization is quantified by projecting the phonon dynamical matrix eigenvectors, shown in Fig. 3b. This is an interesting effect, because typical phonon eigenstates are orthogonal and independent, and to our knowledge, it was only observed in $K_{0.3}MoO_3$ using time-resolved pump-probe spectroscopy (see Ref 56 in the first version). We argue that this is due to strong electron-phonon coupling, which mixes the CDW related phonons. The strong electron-phonon coupling may also be responsible for the absence of phonon softening, as mentioned in the reply to the first comment.

The authors claim that their data shows that the A_{1g} mode survives beyond 60 K, (see, end of page 6) if I look at figure 2c this does not seem to be the case.

The reviewer is absolutely correct that by looking at Fig. 2c it is quite difficult to discern the A_{1g} beyond 60 K. This is due to the strong damping of the amplitude mode when approaching T_{CDW} from below, as seen in many CDW materials (Refs 39–44 in the previous version). That is where the Raman intensity color plot shows its advantage. We emphasized (at the end of Page 6) that in Fig. 2a (instead of Fig. 2c) the existence of the A_{1g} mode can be noted above 60 K. In Line 145 we explained that “It (the A_{1g} mode) is overdamped before disappearing, visualized in the color plot in Fig. 2a as the streak of signal below 100

FIG. R3. A detailed inspection of the A_2 mode. The Raman intensity color plot of (a) the raw data and (b) the data after subtracting the 130 K spectrum. (c) Spectra at 40–90 K (blue) and 120 K (orange) after subtracting the 130 K spectrum. The shaded blue highlights CDW-induced intensity, mainly from the A_2 mode that redshifts and broadens upon warming, but also from the A_1 mode at 45 cm^{-1} which does not shift with temperature.

cm^{-1} between 60–90 K.” Note that this is not an experimental artefact, because we also see this broad streak in a 33.0 nm sample, but not in a 8.3 nm sample when the CDW is destroyed due to degraded sample quality. See comparison in Fig. R2.

To make this more apparent to the readers, we add Fig. R3 to the Supplementary Information. In this figure we show that this mode (i.e. the A_2 mode) can be better resolved by subtracting a temperature independent background from the raw data, taken as the spectrum at 130 K. It is clear that, with such analysis, the A_2 mode can be seen to survive beyond 60 K in both the intensity color plot and the spectra.

The frequency dependences of A_2 and E_3 modes called by the authors CDW amplitude modes seem very similar to those of E_1 – E_4 and A_3 – A_5 . Why the authors focus on these in particular?

The reviewer probably refers to the section “Temperature dependence of Raman modes”, in which when we first described the experimental data, we focused on the A_2 and E_3 modes. The reason we focused on them is that they are the most intense ones among all the observed CDW-induced modes in the A_{1g} and E_{2g} symmetry, respectively, so that their evolution with temperature is the clearest to present to the readers. We used them as examples to illustrate the two types of CDW-induced modes, i.e. the amplitude mode (A_2) and zone-folded mode (E_3), that show different temperature dependences.

The similar temperature dependence of the frequencies for these two modes and other CDW-induced modes (E_1-E_4 and A_3-A_5) can be understood as due to the hybridization of these modes, which makes the zone-folded modes exhibit characters of the amplitude modes (frequency shift and peak broadening). As discussed in the response to the reviewer's second question, the hybridization is an unusual behavior not seen in every CDW material. We have quantified the hybridization using DFT, with results shown in Figs. 3 and 4 in the main text. We think that the hybridization is due to electron-phonon coupling.

The Authors should also mention the following references on collective modes in dichalcogenides observed by Raman: Phys. Rev. Lett. 122, 127001 – Published 27 March 2019, Phys. Rev. B 89, 060503(R) – Published 18 February 2014.

We thank the reviewer for providing these interesting and important references. We have included them in the introduction section. In the Discussion section, we have also used these references to point out that our work may stimulate exploring Higgs mode in the kagome superconductors.

I am waiting for clarification on these points mentioned above, before deciding whether or not I support this work. Regards.

We thank the reviewer again for the helpful comments that prompt us to improve our work. We hope that our response and revisions can properly address the reviewer's questions.

REVIEWERS' COMMENTS

Reviewer #1 (Remarks to the Author):

The authors have pretty much addressed all the points I have raised in my previous report. As a result, I have a more positive opinion of the manuscript. However, there are few details that still need to be addressed more clearly. Below, I refer to them using the numbers in the previous report:

vi) I think there is still some confusion about how one can obtain phonons in structures that do not have optimized forces. In the response to my previous report, the authors state that "intermediate structures the forces on the atoms are nonzero and this is the reason that the intermediate structure still has apparent imaginary modes", which I strongly disagree with: Phonon frequencies depend on the derivatives of the forces, not the forces in the considered structure itself. In principle, one can have a structure with nonzero forces on the atoms, and calculate the first derivative of those forces, and still end up with all positive & real frequencies. What is important is to 1) clarify how the derivatives of the forces are calculated (since the usual scripts/programs used often assume the forces of the starting point is zero, which is clearly not the case here, so more care needs to be taken to calculate those derivatives), and 2) explain that the phonons calculated in these intermediate structures may indeed carry some meaning and illustrate the point, but are not real phonons since they do not correspond to an oscillation around an equilibrium point, or a saddle point or maximum in the energy of the lattice, since the starting structure around which 'phonons' are defined is neither stable, nor metastable, or even an unstable maximum.

viii) The discussion of the uniaxial order in the STM experiments certainly adds to the paper, but is certainly not enough in discussing a possible second transition. In fact, what the STM sees is likely a surface phenomenon only. However, there are many other reports of a second transition in bulk, including elastoresistance (Nie et al. Nature volume 604, pages59–64 (2022)), NMR, as well as multiple muon studies on these materials. At least some of these should be mentioned in the paper, since if the second transition was only seen in STM, it would not have been such a big deal in the first place.

ix) The question of impact and readership: I think this paper is better, more impactful, and of more general interest than most papers in Nature Communications, and as a result, I suggest it for publication after the points above are reconsidered.

Reviewer #2 (Remarks to the Author):

The authors have satisfactorily addressed my questions and comments. I can now recommend publication in Nature Communications.

Reviewer #3 (Remarks to the Author):

Dear Editor,

I read the various answers made by the authors to the referees including mine. I find that the authors have conscientiously answered each of the questions asked by the referees, specifying each point with honesty.

Consequently, I support the publication of this article in Nature Communications

Best regards

Response to Reviewer 1

The authors have pretty much addressed all the points I have raised in my previous report. As a result, I have a more positive opinion of the manuscript. However, there are few details that still need to be addressed more clearly. Below, I refer to them using the numbers in the previous report:

We are glad that the reviewer has a more positive view on our revised manuscript. We appreciate his/her efforts for the second-round review and address the new questions as follows.

vi) I think there is still some confusion about how one can obtain phonons in structures that do not have optimized forces. In the response to my previous report, the authors state that “intermediate structures the forces on the atoms are nonzero and this is the reason that the intermediate structure still has apparent imaginary modes”, which I strongly disagree with: Phonon frequencies depend on the derivatives of the forces, not the forces in the considered structure itself. In principle, one can have a structure with nonzero forces on the atoms, and calculate the first derivative of those forces, and still end up with all positive & real frequencies. What is important is to 1) clarify how the derivatives of the forces are calculated (since the usual scripts/programs used often assume the forces of the starting point is zero, which is clearly not the case here, so more care needs to be taken to calculate those derivatives), and 2) explain that the phonons calculated in these intermediate structures may indeed carry some meaning and illustrate the point, but are not real phonons since they do not correspond to an oscillation around an equilibrium point, or a saddle point or maximum in the energy of the lattice, since the starting structure around which ‘phonons’ are defined is neither stable, nor metastable, or even an unstable maximum.

We thank the reviewer for these helpful comments.

1) We used the “frozen phonon” method to calculate the derivatives of the forces. Specifically, for each intermediate structure we calculated the force differences between the slightly perturbed structure and the unperturbed one to get the correct force constants. The intermediate “phonon spectra” are then calculated based on the correct force constants.

2) Under the harmonic approximation, the existence of the residual forces on the atoms do NOT affect the phonon spectra. This is because the force constants, the second-order derivatives of the potential energy $V = V_0 + \frac{1}{2}kx^2$, does not rely on the forces ($F = -kx$) which are the first-order derivatives of V . However, beyond the harmonic approximation the calculated “phonon spectrum” varies when the intermediate structure varies as a result of phonon-phonon interactions. In this case the physical picture of phonon become illusive, as pointed out by the reviewer. To clarify it, we call it “pseudo phonon spectrum” instead of “phonon spectrum”.

To clarify these points, we made the following changes to the manuscript.

In the section “Nature of CDW-induced modes”, we add “Despite that intermediate structures in Fig. 3 are unstable structures with finite atomic forces, calculated force constants are still valid in the harmonic approximation (see Methods). Corresponding pseudo-phonon bands can provide useful insights to understand the soft mode evolution with respect to CDW distortion.”

In the Methods section, we add “For the pseudo-phonon spectra of the intermediate structures in Fig. 3, we used the same frozen phonon method to calculate force derivatives and obtained force constants, which is valid in the harmonic approximation. Specifically, for each intermediate structure we calculated the force differences between the slightly perturbed structure and the unperturbed one.”

viii) The discussion of the uniaxial order in the STM experiments certainly adds to the paper, but is certainly not enough in discussing a possible second transition. In fact, what the STM sees is likely a surface phenomenon only. However, there are many other reports of a second transition in bulk, including elasto-resistance (Nie et al. Nature volume 604, pages 59–64 (2022)), NMR, as well as multiple muon studies on these materials. At least some of these should be mentioned in the paper, since if the second transition was only seen in STM, it would not have been such a big deal in the first place.

The reviewer is correct that except for the unidirectional order observed by STM, there is now growing evidence for a second low-temperature transition which is of bulk nature. Following the reviewer’s suggestion, we now briefly mention this in the revised version by adding citations to the relevant new studies.

In the paragraph before the Discussion section, we add “A second bulk transition well below T_{CDW} was recently revealed by multiple techniques [53–55], which evaded detection by our Raman measurements, possibly also due to the lack of sufficient sensitivity.” The references 53–55 are representative elasto-resistance, NMR, and muon studies showing evidence for the second transition.

ix) The question of impact and readership: I think this paper is better, more impactful, and of more general interest than most papers in Nature Communications, and as a result, I suggest it for publication after the points above are reconsidered.

We have addressed the above questions. We appreciate the reviewer’s positive comments on the possible impact and general interest of our work.

Response to Reviewer 2

The authors have satisfactorily addressed my questions and comments. I can now recommend publication in Nature Communications.

We are glad that the reviewer is satisfied with our revised version of the manuscript. We thank the reviewer again for taking time to read our manuscript and for helping us to improve our work.

Response to Reviewer 3

Dear Editor, I read the various answers made by the authors to the referees including mine. I find that the authors have conscientiously answered each of the questions asked by the referees, specifying each point with honesty. Consequently, I support the publication of this article in Nature Communications. Best regards.

We thank the reviewer again for his/her thorough review of our manuscript and for the constructive comments. We are glad to know that the reviewer now recommends our work for publication.